# Dual-Activated Nano-Prodrug for Chemo-Photodynamic Combination Therapy of Breast Cancer

**DOI:** 10.3390/ijms232415656

**Published:** 2022-12-10

**Authors:** Ziyao Lu, Gan Xu, Xiaozhen Yang, Shijia Liu, Yang Sun, Li Chen, Qinying Liu, Jianyong Liu

**Affiliations:** 1Fujian Provincial Key Laboratory of Tumor Biotherapy, Clinical Oncology School of Fujian Medical University, Fujian Cancer Hospital, Fuzhou 350014, China; 2Fujian Provincial Key Laboratory of Medical Instrument and Pharmaceutical Technology, College of Biological Science and Technology, Fuzhou University, Fuzhou 350108, China; 3National & Local Joint Biomedical Engineering Research Center on Photodynamic Technologies, College of Chemistry, Fuzhou University, Fuzhou 350108, China; 4Department of Gynecology, Clinical Oncology School of Fujian Medical University, Fujian Cancer Hospital, Fuzhou 350014, China; 5Key Laboratory of Molecule Synthesis and Function Discovery, Fujian Province University, College of Chemistry, Fuzhou University, Fuzhou 350108, China; 6State Key Laboratory of Photocatalysis on Energy and Environment, College of Chemistry, Fuzhou University, Fuzhou 350108, China

**Keywords:** nano-prodrug, photodynamic therapy, reactive oxygen species, activation, combination therapy

## Abstract

Herein, we developed a dual-activated prodrug, BTC, that contains three functional components: a glutathione (GSH)-responsive BODIPY-based photosensitizer with a photoinduced electron transfer (PET) effect between BODIPY and the 2,4-dinitrobenzenesulfonate (DNBS) group, and an ROS-responsive thioketal linker connecting BODIPY and the chemotherapeutic agent camptothecin (CPT). Interestingly, CPT displayed low toxicity because the active site of CPT was modified by the BODIPY-based macrocycle. Additionally, BTC was encapsulated with the amphiphilic polymer DSPE-mPEG_2000_ to improve drug solubility and tumor selectivity. The resulting nano-prodrug passively targeted tumor cells through enhanced permeability and retention (EPR) effects, and then the photosensitizing ability of the BODIPY dye was restored by removing the DNBS group with the high concentration of GSH in tumor cells. Light-triggered ROS from activated BODIPY can not only induce apoptosis or necrosis of tumor cells but also sever the thioketal linker to release CPT, achieving the combination treatment of selective photodynamic therapy and chemotherapy. The antitumor activity of the prodrug has been demonstrated in mouse mammary carcinoma 4T1 and human breast cancer MCF-7 cell lines and 4T1 tumor-bearing mice.

## 1. Introduction

Cancer has become one of the most important causes of death in the past few decades due to invasion and metastasis [1,2]. Although chemotherapy has always been a powerful tool to induce apoptosis or necrosis of cancer cells [3,4], it has limited effects due to its low selectivity and serious systemic toxicity [5]. Therefore, there is an urgent need to develop antitumor drugs with low toxicity and high efficiency.

Photodynamic therapy (PDT) is regarded as one of the most effective treatments for cancer due to its small trauma [6], poor toxicity and high spatiotemporal selectivity [7,8]. PDT contains three critical components, including a photosensitizer (PS), light with a specific wavelength and oxygen to generate reactive oxygen species (ROS), particularly singlet oxygen, to induce cell apoptosis or necrosis [9]. Pioneering studies have confirmed that the combination of PDT and chemotherapy is a promising treatment mode, which not only reduces the dose of the most toxic component but also overcomes single-mode limitations [10,11]. However, most encapsulation or conjugation of PSs and chemotherapeutic agents typically result in drug leakage and low drug toxicity [12,13]. An effective method has been developed to overcome the above shortcomings and improve drug biological activity. Specifically, chemotherapeutic agents are activated by internal (such as hypoxia, high concentrations of GSH, etc.) or external (such as temperature, ROS, etc.) stimuli to release on-demand drugs [14,15]. Youngjae You et al. linked phthalocyanine to PTX using a ROS-sensitive linker. ROS produced after illumination can be used to kill cancer cells as well as cut off the linker to release PTX [16,17]. However, prodrugs can still cause phototoxicity and side effects to normal cells due to their poor selectivity and irregular tumor margins [18,19]. Thus, more practical PSs that only work in tumor tissue through activation by the tumor environment have emerged.

Compared with the normal cell microenvironment, tumor cells have some inimitable pathological signals, such as vascular abnormalities, high concentrations of GSH, acidosis and hypoxia [20,21,22,23]. The PSs lose their photoactivity by connecting to quenchable groups through chemical bonds sensitive to the tumor microenvironment. In the tumor microenvironment, the chemical bonds break, the quenchable effect disappears and the photoactivity of PSs recovers [24,25,26]. There are many types of intermolecular quenching mechanisms, such as photoinduced electron transfer (PET) [27,28], forster resonance energy transfer (FRET) [29,30] and internal charge transfer (ICT) [31,32]. Akkaya et al. designed a GSH-activated BODIPY based on the PET effect between the 2,4-dinitrobenzenesulfonate (DNBS) group and the parent nucleus of BODIPY [33]. Zheng et al. designed an MMP-7-responsive PS in which porphyrin and a quenching agent (BHQ3) could cause the FRET effect [34]. Zhao et al. designed an activated PS based on the ICT effect between the DNBS group and iodinated BODIPY [35]. Although environmentally responsive PSs avoid damage to normal tissues, nontargeting agents still lead to low PDT efficiency [36,37]. To solve the above drawback, specific chemical bonds are usually added to PSs to connect polypeptides, vitamins, antibodies and small molecules, which increase tumor targeting [38,39,40,41,42]. Alternatively, taking advantage of the large space between vascular endothelial cells at the tumor, nanoparticles loaded into PSs also enter the tumor tissue by virtue of the enhanced permeability and retention (EPR) effect [43,44,45,46].

In this study, we designed a prodrug that combines passive targeting and stepwise activation. First, the fluorescence of BODIPY was quenched due to the PET effect existing between BODIPY and the DNBS group. In an environment of high concentration GSH, the sulfhydryl group of GSH and the nitro sulfonate structure undergo a nucleophilic substitution reaction which leaves the 2,4-dinitrobenzenesulfonyl moiety, the PET effect is inhibited and the fluorescence of BODIPY is restored. Then, CPT is attached to BODIPY by means of a ROS-sensitive linker. The fluorescence of CPT is quenched by BODIPY derivatives, accounting for FRET, and the low toxicity of CPT is displayed because the active site of CPT is modified by a BODIPY-based macrocycle. Meanwhile, nanoparticles are coated with the amphiphilic polymer DSPE-mPEG_2000_ to improve drug solubility and tumor selectivity. As demonstrated in Figure 1, after entering the tumor through passive targeting, the prodrug is activated by a high concentration of GSH in the tumor, and ROS is produced under light stimulation to damage tumor cells. Meanwhile, ROS ruptures the TK linker between BODIPY and CPT, releasing CPT to achieve the combination treatment of selective PDT and chemotherapy.

## 2. Results and Discussion

### 2.1. Molecular Design and Synthesis

In this paper, we rationally designed GSH and a ROS-responsive prodrug that consists of the DNBS group, BODIPY-based PS, a ROS-cleavable thioketal linker and CPT. As is known, the BODIPY-based fluorescent dye is a hopeful fluorescent chromophore, which is mainly due to its strong absorption in the visible and near infrared region (NIR), and robust photo-stability. Interestingly, its spectral properties can be significantly tuned by reasonable chemical modification. Figure 2 depicts the synthetic route for the BTC. First, the distyryl-substituted BODIPY **1** was modified with the 3,4-dinitrobenzenesulfonyl chloride to produce GSH-responsive BODIPY **2**. Owing to photo-induced electron transfer (PET), GSH-responsive BODIPY **2** had a relatively weak fluorescence emission and a red-shift absorption, which is beneficial to the PDT of deep tissues. Then, BODIPY **2** was clicked with covalent CPT **3a** containing a ROS-sensitive linkage to attain BTC. For comparison, the reference compounds with ROS-insensitive carbon chains, BCC was also prepared in a similar procedure. In brief, BTC and BCC were nontoxic to healthy cells with or without light. In contrast, BCC only restored phototoxicity in tumor cells with a laser. However, BTC restored phototoxicity and chemotoxicity. The synthesis process and characterization in the Appendix A demonstrate that BTC and BCC were successfully obtained (Appendix A).

### 2.2. Photophysical and Photochemical Properties

#### 2.2.1. Photophysical Properties

The electronic absorption spectra of BTC, BCC, BODIPY **2** and CPT were measured in dimethylsulfoxide (DMSO). As shown in Appendix A, BTC, BCC and BODIPY **2** showed a strong Q band with a maximum absorption wavelength of 683 nm. It was found that the maximum absorption wavelength of BODIPY **2** was redshifted by 10 nm compared with that of BODIPY **1** (Appendix A), which may have been due to the strong electron-absorbing effect of introducing the DNBS group. In addition, the absorption of BTC and BCC were the superposition of CPT and BODIPY **2**, which indicated that the introduction of CPT negligibly affected the electron absorption of BTC and BCC (Appendix A). When BODIPY **1** was selectively excited at 610 nm, BODIPY **1** exhibited considerable fluorescence emission. The fluorescence quantum yield (ΦF) of BODIPY **1** was determined to be 0.319 relative to unsubstituted zinc (II) phthalocyanine (ZnPc). However, BODIPY **2**, BCC and BTC were virtually nonfluorescent. This change was attributed to the conjugation with the DNBS group. Their fluorescence quantum yields were estimated to be 0.081, 0.076, 0.078, respectively (Figure 1a and Appendix A). This result demonstrated that the DNBS moiety could effectively reduce the fluorescence emission.

#### 2.2.2. GSH-Responsive Behavior

The intracellular concentration of GSH is approximately one thousand times that of extracellular GSH. Here, we used concentrations of 2.5 μM and 2.5 mM GSH to mimic the inside and outside of the tumor cells, respectively. As can be seen from Figure 1b and Appendix A, the fluorescence emission intensity of BCC and BTC at 718 nm showed no significant change after two hours of incubation with 2.5 μM GSH. However, when they were incubated with 2.5 mM GSH (Figure 1b and Appendix A), the fluorescence emission intensity steadily increased with the extension of incubation time and finally stabilized, and the fluorescence intensity of BTC and BCC at 718 nm increased remarkably, by approximately 4-fold, compared with incubating with 2.5 μM GSH for 2 h, showing that the DNBS group was susceptible to thiol-mediated cleavage in a millimolar concentration of GSH, and the recovery degree of PS fluorescence was positively correlated with the concentration of GSH and incubation time. This also means that BTC and BCC can remain in a quenched state in the blood circulation; however, once they enter the tumor cells, they can be activated rapidly by GSH inside the tumor cells.

As expected, the singlet oxygen (^1^O_2_) generation efficiency of BTC and BCC also responded similarly. The trapping agent of singlet oxygen, 1,3-diphenyliso-benzofuran (DPBF), is the best choice for singlet oxygen detection due to its high stability and sensitivity. When the ^1^O_2_ reacts with DPBF in solution, the conjugated structure of DPBF will be destroyed and the absorption of DPBF at 415 nm will be reduced. Therefore, the ^1^O_2_ production capacity of the compound was indirectly reflected by observing the decrease rate of DPBF absorption at 415 nm. As depicted in Appendix A, there were no considerable absorption changes for DPBF at 415 nm in the presence of BTC and BCC under light (660 nm, 1 mW/cm^2^). The results indicated that both BTC and BCC did not effectively generate singlet oxygen. Under 2.5 μM GSH, there was essentially unchanged absorbance of DPBF at 415 nm under irradiation at 660 nm (Figure 1c and Appendix A), which indicated that BTC and BCC could not effectively generate singlet oxygen after coculturing for 120 min with 2.5 μM GSH. However, after incubating these two compounds with 2.5 mM GSH, the absorbance of DPBF at 415 nm decreased remarkably (Figure 1c and Appendix A), indicating that these two compounds obviously boosted the photo-degradation of DPBF. All the above findings revealed that the intramolecular PET process in BTC and BCC disappeared with the high concentration of GSH, increasing the fluorescence emission and singlet oxygen generation efficiency.

#### 2.2.3. ROS-Triggered CPT Release

The first step was to confirm the intramolecular FRET process in BTC and BCC. The fluorescence of BTC or BCC was compared with that of CPT under the same conditions. Upon excitation at 370 nm, the fluorescence of CPT was strong at 430 nm (Appendix A). By contrast, we found that the fluorescence intensity of BTC and BCC decreased dramatically at 430 nm, and a new fluorescence peak appeared at 724 nm (Appendix A). This phenomenon accounted for the FRET effect between BODIPY units and CPT, such that the fluorescence of CPT was quenched by BODIPY units. The next step was to evaluate the CPT moiety release in the prodrug BTC by monitoring the fluorescence changes at 430 nm under different conditions. According to Figure 1d and Appendix A, after the treatment of the solution of BTC with laser light (660 nm, 30 mW/cm^2^), the fluorescence intensity of CPT hardly changed, indicating the CPT moiety was barely released. After pretreating with 2.5 μM of GSH for 120 min, the fluorescence emission of CPT moiety from BTC was insignificant under laser irradiation (660 nm, 30 mW/cm^2^) (Appendix A). By contrast, after treatment with 2.5 mM GSH for 120 min (Figure 1d and Appendix A), the fluorescence of CPT in the solution increased significantly and reached a plateau under light (660 nm, 30 mW/cm^2^). Due to the fluorescence of the CPT moiety in BTC being in the “off” state through the FRET process, the increased fluorescence intensity at 430 nm was attributed to the released CPT from BTC. Furthermore, the fluorescence of CPT could not enhance the reference BCC upon irradiation after the treatment with 2.5 mM GSH (Figure 1d and Appendix A).

### 2.3. Preparation and Characterization of BTC NPs and BCC NPs

Since BTC and BCC are water-insoluble, we wrapped BTC and BCC into soluble nanoparticles by using the amphiphilic polymer DSPE-mPEG_2000_, which can increase drug solubility but can also improve tumor targeting through the EPR effect. The hydrodynamic diameters of BTC NPs and BCC NPs were 99.7 nm and 112.5 nm, respectively, and the polydispersity index (PDI) of BTC NPs and BCC NPs were approximately 0.2 (Figure 2a and Appendix A), showing a promising distribution in solution. Transmission electron microscopy (TEM) results showed that the BTC NPs and BCC NPs had uniform and monodispersed spherical shapes. Additionally, the loading content and loading efficiency of BTC (BCC) in the BTC NPs (BCC NPs) were 1.5% (1.3%), and 85% (80%), respectively. The BTC NPs and the BCC NPs also showed superior stability in an aqueous solution, and no precipitation or significant changes in average size were observed within 7 days (Figure 2b and Appendix A), which guaranteed their potential application in vivo. We found that the maximum absorption wavelength of the BTC NPs and BCC NPs was lower than those of BTC and BCC, which confirmed the BTC NPs and BCC NPs have aggregation (Figure 2c and Appendix A). Upon excitation at 640 nm, both NPs emitted a negligible fluorescence (Figure 2d and Appendix A), which indicated that aggregation favored quenching the fluorescence.

### 2.4. In Vitro Assessments

#### 2.4.1. GSH-Responsive Intracellular Fluorescence

The activation effect of BTC NPs in mouse mammary carcinoma 4T1 and human breast cancer MCF-7 cell lines was also monitored to estimate the intracellular fluorescence of BDP units via confocal laser scanning microscopy (CLSM). According to Figure 3a,c and Appendix A, both the BTC NPs and BCC NPs showed bright intracellular fluorescence in treated 4T1 and MCF-7 cells, indicating that the nanoparticles could be activated by GSH in tumor cells. In contrast, the intracellular fluorescence intensity of the BTC NPs and BCC NPs decreased significantly after the tumor cells were pretreated with L-buthionine sulfoximine (BSO) to deplete intracellular GSH [47], which indicated that GSH inside the tumor cells should be the key to activating the fluorescence emission of the BTC NPs and BCC NPs.

#### 2.4.2. Intracellular ROS Level

Since ROS produced by PSs can directly kill tumor cells, the ROS generation ability of PSs in tumor cells is extremely important. Here, the release of CPT was also controlled by the ROS. Next, we investigated ROS generation in 4T1 and MCF-7 cells using 2,7-dichlorofluorescein yellow diacetate (DCFH-DA) as the ROS indicator. After DCFH-DA enters cells through diffusion, DCFH is rapidly oxidized by intracellular ROS and then generates DCF with strong fluorescence, which can be detected by flow cytometry. There was negligible fluorescence in 4T1 (Figure 3b,d) and MCF-7 cells (Appendix A) treated with PBS with or without light at 660 nm, indicating that the content of endogenous ROS in cells was relatively low. When the BTC NPs- and BCC NPs-treated cells were stimulated with a laser at 660 nm, the fluorescence was markedly enhanced and was 5-fold higher than that without laser stimulation. The results implied that nanoparticles could be activated by intracellular GSH and produced large amounts of ROS under light. We also investigated whether ROS could still be produced by the nanoparticles when intracellular GSH was depleted. As the results showed, when cells were pretreated with BSO followed by the BTC NPs and BCC NPs, the intracellular fluorescence of DCF was significantly weakened, demonstrating that BSO reduced the intracellular concentration of GSH, leading the prodrug to inactivate.

#### 2.4.3. Intracellular CPT Release

We further evaluated the release efficiency of CPT from the BTC NPs triggered by light at the cellular level by CLSM. After treatment with BTC NPs for 24 h, there was only negligible blue fluorescence in the CPT channel in 4T1 (Figure 4a,c,d) and MCF-7 cells (Appendix A). However, the fluorescence of CPT was significantly enhanced when illuminated at 660 nm for 2 min, demonstrating the release of CPT from the BTC NPs under light. Moreover, regardless of the light, the fluorescence of CPT from the BCC NPs was very minimal due to the absence of an ROS-cleavable linker. Notably, when the cells were treated with the BTC NPs or BCC NPs, the red fluorescence in the BODIPY channel of the laser group was similar to that of the no-laser group.

#### 2.4.4. Cytotoxicity Assays

To explore the killing activity of nanoparticles on tumor cells directly, the cytotoxicities of the BTC NPs and BCC NPs were investigated in 4T1 and MCF-7 cells by methyl thiazolyl tetrazolium (MTT) assays. In Figure 4e and Appendix A, the BTC NPs and BCC NPs exhibited negligible dark toxicity, even at a concentration of 10 μM. The results indicated the cytotoxicity of CPT was largely inhibited by conjugating with BODIPY-based PSs. However, following laser exposure, the half maximal inhibitory concentrations (IC50) values for the BTC NPs were decreased to 0.50 μM for 4T1 cells and 0.63 μM for MCF-7 cells, which were lower than the BCC NPs (1.6 and 1.5 μM, respectively). The cytotoxicity differences between the BTC NPs and the BCC NPs were probably because the BTC NPs killed the tumor cells by ROS and ROS-triggered release of CPT in a combination treatment of PDT and chemotherapy, while the BCC NPs damaged the tumor cells only by ROS via PDT treatment. Furthermore, we detected the cell apoptosis induced by the BTC NPs and the BCC NPs by an Annexin V-FITC/PI Apoptosis Kit. As shown in Figure 4b, both the BTC NPs and BCC NPs induced obvious apoptosis under light conditions, but the apoptosis rate induced by the BTC NPs was higher than that induced by the BCC NPs, which suggested that the BTC NPs could kill tumor cells more efficiently through the combination of PDT and chemotherapy.

### 2.5. In Vivo Studies

The biodistribution of the BTC NPs in tumor tissues was evaluated in 4T1 tumor-bearing BALB/c mice. Because BODIPY-based PSs could emit near-infrared (NIR) fluorescence, the time-dependent accumulation process of the BTC NPs was directly observed by fluorescence molecular tomography (FMT). As shown in Figure 5a, there was a fluorescence signal for mice after tail vein injection with the BCC NPs and BTC NPs for 4 h. The fluorescence signal increased first and then decreased, and the maximum accumulation time of BCC NPs and BTC NPs was 12 h. This was because of the EPR effect of the nano-prodrug which caused pronounced tumor enrichment of BCC NPs and BTC NPs; the nano-prodrug was further activated by high concentrations of GSH in the tumor tissues. Furthermore, BCC and BTC displayed weak fluorescence signals at the tumor sites (Appendix A), which was due to poor tumor accumulation. We further collected tumor tissue and other main organs to analyze the prodrug distribution at the maximum accumulation time and found that the concentrations of the BCC NPs and BTC NPs in tumor tissue were much higher than that in other main organs (Figure 5b), suggesting that the nano-prodrug could target tumor tissues. Additionally, similar results for the fluorescence image were observed in BTC and BCC (Appendix A).

Furthermore, we investigated the antitumor effect of the BTC NPs and BCC NPs with or without a light trigger in vivo. Thirty female BALB/c mice bearing 4T1 tumors were blindly separated into six groups treated with saline, saline plus light, BCC NPs, BCC NPs plus light, BTC NPs, and BTC NPs plus light. As shown in Figure 6a, tumors grew rapidly in mice treated with saline with or without laser irradiation. A similar tumor growth tendency was displayed for mice treated with the BCC NPs or the BTC NPs but without light, showing negligible dark toxicity. However, both BTC NPs plus light and BCC NPs plus light exhibited satisfactory antitumor effects. More interestingly, BTC NPs plus light displayed better antitumor efficacy than BCC NPs plus light, revealing the combined therapeutic effect of photodynamic therapy and chemotherapy. The tumor weight (Figure 6b) and photographs (Figure 6d) also confirmed the above results. We further examined the antitumor effect of the BTC NPs by analyzing hematoxylin and eosin (H&E)-stained tumor tissues. As shown in Figure 6e, there was almost no cell necrosis or apoptosis in the group treated with saline, saline plus light, BCC NPs or BTC NPs, while a large number of shrinking and nuclear pyknosis cells appeared in the other two groups of BCC NPs plus light and BTC NPs plus light. Moreover, tumor cell proliferation seemed to be inhibited significantly in the BTC NPs plus light group.

Finally, the safety of the BTC NPs and BCC NPs were evaluated by observing the body weight change of mice under treatment and analyzing H&E-stained main organs (heart, liver, spleen, lung and kidney). The body weight of mice after BTC NPs plus light and BCC NPs plus light treatment remained stable (Figure 6c). Additionally, there were no pathological changes in the main H&E-stained organs in the BTC NPs plus light and BCC NPs plus light group (Appendix A), suggesting that no obvious toxicity or side effects were observed. Overall, the above results proved that BTC NPs had superior biocompatibility and outstanding antitumor effects with light.

## 3. Materials and Methods

### 3.1. General

The purification of solvent, instrumentation, photophysical and photochemical investigations are described in the Appendix A. Compounds **1**, **2**, **3a** and **3b** were prepared, as previously reported [48].

#### 3.1.1. Synthesis of BTC

A solution of **2** (0.16 g, 0.12 mmol) and **3a** (21 g, 31 μmol) in dichloromethane (6 mL) was added to a mixture of CuSO_4_·5H_2_O (15 mg, 60 μmol) and sodium ascorbate (30 mg, 0.15 mmol) in water (0.5 mL). Then, 0.5 mL of ethanol was added. The resulting mixture was stirred overnight under N_2_, poured into saturated sodium chloride and extracted with dichloromethane twice (2 × 15 mL). The crude product was further purified by column chromatography on silica gel using dichloromethane/MeOH (50/1, *v*/*v*) as the eluent to produce BTC as a dark green solid. (39 mg, 65%). ^1^H NMR (400 MHz, CDCl_3_): δ = 8.71 (s, 1 H), 8.55 (d, J = 8.4 Hz, 1 H), 8.37 (s, 2 H), 8.25 (d, J = 8.8 Hz, 1 H), 8.18 (d, J = 8.8 Hz, 2 H), 8.00 (d, J =16.4 Hz, 2 H), 7.90 (d, J = 8. 4 Hz, 2 H), 7.88 (s, 2 H), 7.80 (t, J = 7.6 Hz, 2 H), 7.63 (t, J = 7.6 Hz, 2 H), 7.52 (d, J = 16.4 Hz, 2 H), 7.44 (d, J = 8.0 Hz, 2 H), 7.39 (d, J = 7.6 Hz, 2 H), 7.31 (s, 2 H), 7.20 (d, J = 8.4 Hz, 4 H), 6.90 (d, J = 8.4 Hz, 2 H), 5.67 (d, J = 17.2 Hz, 2 H), 5.36 (d, J = 16.8 Hz, 2 H), 5.28 (s, 4 H), 5.24 (s, 4 H), 4.57 (t, J = 4.8 Hz, 4 H), 4.46 (t, J = 5.2 Hz, 4 H), 4.27–4.20 (m, 8 H), 4.15 (t, J = 6.8 Hz, 4 H), 3.89 (t, J = 4.8 Hz, 4 H), 3.74 (t, J = 4.4 Hz, 4 H), 3.67–3.62 (m, 8 H), 3.52 (t, J = 4.0 Hz, 4 H), 3.34 (s, 6 H), 2.85 (t, J = 6.8 Hz, 4 H), 2.78 (t, J = 6.8 Hz, 4 H), 2.29–2.09 (m, 4 H), 1.50 (d, J = 9.2 Hz, 12 H), 1.34 (s, 6 H), 0.98 (t, J = 7.2 Hz, 6 H). ^13^C NMR (100.6 MHz): δ 167.33, 157.21, 154.26, 153.44, 152.26, 151.07, 150.89, 149.58, 148.93, 148.76, 148.32, 148.06, 146.52, 145.62, 144.07, 140.38, 138.88, 136.10, 135.00, 133.94, 132.96, 131.86, 131.32, 130.79, 130.67, 130.11, 129.52, 128.45, 128.29, 128.17, 128.02, 126.63, 124,27, 123,27, 122.42, 120.63, 120.11, 116.17, 114.76, 113,99, 110.56, 95.90, 77.98, 71.89, 70.76, 70.61, 70.47, 69.57, 68.59, 67.51, 67.06, 67.04, 65.73, 63.53, 58.97, 56.57, 50.03, 48.88, 31.83, 30.87, 30.85, 28.78, 28.74, 13.87, 7.67. HRMS-ESI (m/z): [M+H]^+^ calced for C_121_H_125_BBr_2_F_2_N_14_O_35_S_5_, 2704.5549, found 2704.5558.

#### 3.1.2. Synthesis of BCC

Similarly, **2** (0.16 g, 0.12 mmol) was treated with **3b** (21 g, 31 μmol), CuSO_4_·5H_2_O (15 mg, 60 μmol) and sodium ascorbate (30 mg, 0.15 mmol) to produce BCC as a dark green solid. (39 mg, 71%). ^1^H NMR (400 MHz, CDCl_3_): δ = 8.70 (s, 1 H), 8.53 (d, J = 8.4 Hz, 1 H), 8.38 (s, 2 H), 8.21 (d, J = 10.0 Hz, 1 H), 8.19 (d, J = 8.8 Hz, 2 H), 8.00 (d, J =16.4 Hz, 2 H), 7.91 (d, J = 8.4 Hz, 2 H), 7.88 (s, 2 H), 7.80 (t, J = 7.6 Hz, 2 H), 7.64 (t, J = 7.6 Hz, 2 H), 7.51 (d, J = 16.8 Hz, 2 H), 7.43 (d, J = 8.0 Hz, 2 H), 7.37 (d, J = 8.0 Hz, 2 H), 7.32 (s, 2 H), 7.21 (d, J = 6.4 Hz, 4 H), 6.92 (d, J = 8.8 Hz, 2 H), 5.67 (d, J = 17.2 Hz, 2 H), 5.37 (d, J = 17.2 Hz, 2 H), 5.29 (s, 4 H), 5.25 (s, 4 H), 4.60 (t, J = 5.2 Hz, 4 H), 4.47 (t, J = 5.2 Hz, 4 H), 4.22 (t, J = 4.8 Hz, 4 H), 4.12–4.04 (m, 4 H), 4.00 (t, J = 6.8 Hz, 4 H), 3.90 (t, J = 4.8 Hz, 4 H), 3.75 (t, J = 4.8 Hz, 4 H), 3.66 (t, J = 4.8 Hz, 4 H), 3.63 (t, J = 4.8 Hz, 4 H), 3.52 (t, J = 4.4 Hz, 4 H), 3.35 (s, 6 H), 2.29–2.11 (m, 4 H), 1.66–1.54 (m, 8 H), 1.37–1.33 (m, 8 H), 1.28 (s, 6 H),0.99 (t, J = 7.6 Hz, 6 H). ^13^CNMR (100.6 MHz): δ 167.47, 157.23, 155.26, 154.54, 153.77, 152.24, 151.06, 150.86, 149.58, 148.94, 148.74, 148.35, 148.05, 146.42, 145.78, 144.11, 138.90, 136.08, 135.00, 133.92, 132.97, 131.84, 131.35, 130.76, 130.71, 130.13, 129.47, 128.51, 128.28, 128.18, 128.04, 126.60, 124.20, 123.27, 122.42, 120.61, 120.16, 116.19, 114.74, 113.98, 110.55, 95.95, 77.68, 71.88, 70.75, 70.60, 70.46, 69.56, 68.89, 68.58, 68.33, 67.02, 65.46, 63.54, 58.96, 50.00, 48.95, 31.86, 28.36, 28.32, 25.18, 13.85, 7.65. HRMS-ESI (m/z): [M+H]^+^ calced for C_119_H_121_BBr_2_F_2_N_14_O_35_S, 2548.6353, found 2548.6357.

### 3.2. Photophysics and Photochemistry Investigations

#### 3.2.1. GSH-Responsive Fluorescence Emission

GSH was dissolved in deionized water to obtain 0.5 M stock solution. BTC and BCC were dissolved in DMSO to produce 2 mM stock solutions. The samples of BTC (or BCC) (2.5 μM) with various concentrations of GSH (0 μM, 2.5 μM, 2.5 mM) in a mixture of DMSO and PBS (*v*/*v*, 3:1) were then obtained. There were four groups: (1) BTC + 2.5 μM GSH; (2) BTC + 2.5 mM; (3) BCC + 2.5 μM GSH; and (4) BTC + 2.5 mM. The fluorescence emission spectra of the sample solutions were determined continuously for 2 h under excitation at 640 nm.

#### 3.2.2. Intramolecular FRET Process Evaluation

CPT, **2**, BTC and BCC were dissolved in DMSO to produce a 1 mM solution, which was then diluted to 2 μM with a mixture of DMSO and PBS (*v*/*v*, 3:1). The fluorescence emission spectrum of these solutions was recorded at 380–900 nm (λ_ex_ = 370 nm).

#### 3.2.3. Photoinduced CPT Release

BTC and BCC were dissolved in DMSO to produce 2 mM stock solutions. GSH was dissolved in deionized water to obtain 0.5 M stock solution. BTC (2.5 μM) and GSH (0 μM, 2.5 μM and 2.5 mM) were prepared in a mixture of DMSO and PBS (*v*/*v*, 3:1) and cocultured at room temperature for 2 h. BCC was directly incubated with GSH (2.5 mM) at room temperature for 2 h. Then, the sample solution was irradiated by a 660 nm laser with a power of 30 mW/cm^2^ for 120 min. The release of CPT was recorded by monitoring the fluorescence changes at 430 nm (λ_ex_ = 370 nm) with illumination time.

### 3.3. Preparation and Characterization of BTC NPs and BCC NPs

#### 3.3.1. Preparation of BTC NPs and BCC NPs

In short, 25 mg DSPE-mPEG_2000_ was dissolved in 10 mL water and ultrasonicated for 5 min at room temperature. A solution of BTC (or BCC) (1 mM) in 100 μL DMSO was added to the DSPE-mPEG_2000_ aqueous solution by dropwise addition and stirred overnight at room temperature to stabilize and create uniform nanoparticles.

#### 3.3.2. Characterization of BTC NPs and BCC NPs

The hydrodynamic diameters of the BTC NPs and BCC NPs were recorded by dynamic light scanning (DLS). The morphology of the BTC NPs and BCC NPs were determined by transmission electron microscopy (TEM). The stability of the BTC NPs and BCC NPs were recorded by detecting the changes in size through DLS over 7 days in phosphate-buffered saline (PBS) solution (10 μM, pH 7.4) at 37 °C.

### 3.4. In Vitro Studies

#### 3.4.1. GSH-Responsive Intracellular Fluorescence

Buthionine sulfoximine (BSO) can inhibit GSH synthesis in cells. The 4T1 and MCF-7 cells (1 × 10^5^) in 1 mL of Dulbecco’s modified Eagle’s medium (DMEM) were seeded and cultured in a confocal dish and incubated overnight. Then, the cells were continued to incubate with BSO (10 mM) in 1 mL DMEM medium for 12 h. Free DMEM without BSO was added to another group. After removing the old medium and rinsing three times with PBS, 1 mL DMEM medium bearing the BTC NPs and BCC NPs (5 μM) was added and incubated for 24 h. The cells were washed three times with PBS, serum-free medium was added to the dishes and intracellular fluorescence images were obtained by confocal laser scanning microscopy (CLSM). The BODIPY unit was excited at 633 nm and collected at 650–800 nm.

#### 3.4.2. Intracellular ROS Measurement

Intracellular ROS levels of the BTC NPs and BCC NPs were detected by a DCFH-DA probe. First, 4T1 and MCF-7 cells (1 × 10^5^) were seeded in a 12-well plate and incubated overnight at 37 °C with 5% CO_2_. Then, the cells were incubated with BSO (10 mM) in 1 mL DMEM medium for 12 h. Free DMEM medium without BSO was added to another group, after removing the old DMEM medium and rinsing three times with PBS, 1 mL DMEM medium containing BTC NPs (or BCC NPs) (2 μM) was added and fresh DMEM was added as a control. After incubation for 24 h, the old DMEM medium was removed, and the plates were rinsed thrice with PBS. The 1 mL DMEM medium bearing DCFH-DA (10 μM) was added to each well and incubated for another 1 h. After washing three times with PBS, fresh DMEM without serum and phenol red was added and the cells were placed under 660 nm laser irradiation at 20 mW/cm^2^ for 5 min. Moreover, flow cytometry was used to collect quantitative data on the intracellular ROS level.

#### 3.4.3. Intracellular CPT Release

First, 4T1 and MCF-7 cells (1 × 10^5^) in 1 mL of DMEM were seeded and cultured in a confocal dish and incubated overnight. After washing three times with PBS, 1 mL DMEM medium of the BTC NPs (or BCC NPs) (5 μM) was added to the dish and incubated for 24 h. Meanwhile, the old medium was removed and rinsed three times with PBS. Then, 1 mL serum-free medium was added to the dishes. Subsequently, the dishes were irradiated with LED irradiation (660 nm, 20 mW/cm^2^) for 2 min, incubated for an additional 15 min and finally imaged by CLSM. The CPT was excited at 405 nm and detected at 425–475 nm and the BODIPY unit was excited at 633 nm and collected at 650–800 nm.

#### 3.4.4. Photocytotoxicity Assay

MTT can detect the cytotoxicity of each compound. In short, 5000 4T1 and MCF-7 cells in 100 μL were seeded into 96-well plates and cultured overnight. Thereafter, the old medium was removed, and the cells were cultured with free medium containing BTC NPs (or BCC NPs) at different concentrations. After 24 h of incubation, the cells were washed three times with PBS and the new DMEM medium was added. The cells were irradiated with or without a 660 nm LED lamp (20 mW/cm^2^) for 5 min and then cultured for another 12 h. Afterward, 10 μL of MTT (5 mg/mL) was added to each well and cultured for 4 h. Finally, the media was discarded and 100 μL of DMSO was added to each well. The optical density (OD) was measured at 570 nm by a microplate reader. The larger the OD value is, the stronger the cell activity. Cell viability = (OD _sample_/OD _control_) × 100%.

#### 3.4.5. Cell Apoptosis Assay

The 4T1 and MCF-7 cells were seeded into 6 wells at a density of 1 × 10^5^ cells per well and incubated overnight. Then, BTC NPs (or BCC NPs) (5 μM) was added for an additional 24 h. The cells were treated with a 660 nm LED lamp at 20 mW/cm^2^ for 5 min and incubated for another 12 h. Meanwhile, the cells were collected, washed three times with PBS, and stained with PI (5 μL) and Annexin V-FITC (15 μL) solutions for 15 min based on the guidelines of the apoptosis kit (Beyotime Biotechnology, C1062L, Shanghai, China). Finally, the cells were placed on ice and detected by flow cytometry.

### 3.5. In Vivo Studies

All BALB/c mice (female, 20–25 g) were purchased from the Wushi Laboratory Animal Services Centre (Fuzhou, China). All experiments were performed according to the Institutional Animal Care and Treatment Committee of Fuzhou University. The healthy 4T1 cells were digested, centrifuged, washed twice with PBS and then collected in centrifugal tubes. Thereafter, 4T1 cells were subcutaneously injected at a density of 2 × 10^6^ cells (100 μL) per mouse on the right side of the back. After the average tumor size reached a certain volume (approximately 80 mm^3^), all animal experiments were performed.

#### 3.5.1. In Vivo Fluorescence Imaging Study

A saline solution of BTC NPs, BCC NPs, BTC and BCC (2 mg/kg) were prepared, and 25 mice were randomly assigned to five groups (BCC NPs, BTC NPs, BCC, BTC and control). Each mouse was injected with 100 μL of solution via the tail vein. The changes in drug fluorescence in mice were observed by using the PerkinElmer FMT2500LX imaging system at different time points (0, 4, 6, 12, 24, 36) after injection. At the time point of maximum concentration of drugs in tumor tissues, the experimental mice were euthanized, and the heart, liver, spleen, lung and kidney were removed. A small animal imager was used to compare the drug distribution in different tissues of each group.

#### 3.5.2. In Vivo Antitumor Assay

Thirty mice were randomly assigned to six groups: (1) saline; (2) saline plus laser; (3) BCC NPs; (4) BCC NPs plus laser; (5) BTC NPs; and (6) BTC NPs plus laser. Each group was injected with 100 μL solution via the tail vein. Then, the mice in the (2), (4) and (6) groups were irradiated with a 660 nm laser at a power of 300 mW/cm^2^ for 10 min. After light treatment, the mice were weighed, and the tumor volume was measured at regular times every day for 15 days.

After treatment, a mouse was randomly selected from each group and killed by the cervical dislocation method. The tumor, heart, liver, spleen, lung and kidney were removed and immersed in 4% paraformaldehyde solution for fixation for 24 h. Paraffin-wrapped sections were prepared for H&E staining. The histological morphology of each tissue section was observed and photographed under a light microscope.

## 4. Conclusions

In conclusion, we designed and synthesized a GSH-activatable and ROS-responsive nano-prodrug, BTC, to attain the target PDT–chemo treatment for breast cancer. The BTC NPs were dormant without chemotoxicity and phototoxicity in healthy tissues. After the BTC NPs entered the 4T1 and MCF-7 tumor cells, the DNBS group was removed and the BODIPY derivative was activated by the high concentration of GSH. Subsequently, BTC NPs generated ROS after irradiation, and then the ROS cleaved the ROS-cleavable TK linkage to realize on-demand CPT release. The IC_50_ values for the BTC NPs were 0.50 μM toward 4T1 cells and 0.63 μM against MCF-7 cells, showing strong photocytotoxicities to tumor cells. The BTC NPs have been proven to have efficient tumor-targeting and tumor-suppressive effects without obvious toxicity and side effects in 4T1 tumor-bearing mice, which is of great significance for cancer treatment.

## Data Availability

Not applicable.

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
