# Peer review of "Dual-Activated Nano-Prodrug for Chemo-Photodynamic Combination Therapy of Breast Cancer"

_ijms, 2022, doi:10.3390/ijms232415656_

Round 1

Reviewer 1 Report

In the manuscript entitled “Dual-activated nanoprodrug for chemo-photodynamic combination therapy of breast cancer” the authors report on the synthesis and properties of an innovative prodrug for the treatment of cancer designed by combining the photoluminescent properties of a BODIPY unit with the antitumoral activity of camptothecin by means of a ROS sensitive spacer.

The rationale of this approach relies on the synergic action of the photodynamic therapy and of the chemotherapy, triggered by the high level of reactive oxygen species and glutathione found in the tumoral tissue that can activate the prodrug, and potentiated by the specific nanoformulation ensuing for a passive targeting process.

The topic is very hot as denoted by the state of the art proposed in the introduction section supported by a selection of relevant and recent literature. The rationale at the basis of the design of the prodrug is well discussed and the great mole of experiments is well-conceived.

Unfortunately, despite the great premises, the quite interesting results have not been discussed in an appropriate manner and the reported graphs are often inaccurate as well as the corresponding legends. The main outcomes have not been adequately valorized, even in the conclusion section where the most important results have not been fully stressed.

The experimental section is very imprecise and some details are often lacking, enabling for a correct reproduction of the setup.

Finally, an extensive English editing is strongly committed.

For these reasons, the manuscript in the present form is not suitable for the publication in the International Journal of Molecular Sciences. However, due to the very interesting results and to the innovative design of the prodrug and its formulation, I strongly recommend that the authors revise the entire manuscript and reconsider to submit it again.

Reviewer 2 Report

Overview and general recommendation:

This paper describes that a dual activatable prodrug BTC is developed and the photophysical&photochemical properties are defined. The authors also describe the characterization of BTC NPs. The authors perform in vitro assay and the results show that light-triggered ROS can induce apoptosis or necrosis of tumor cells and the release of CPT. The in vivo assay shows that tumor cell proliferation and tumor size can be inhibited by BTC NPs+lights.

Overall the manuscript is OK. The authors performed detailed background research and the research is designed rationally. The result is good enough to support their conclusions. Figures are organized in a proper way. I suggest the authors to add some data to show the enhanced permeability. I also suggest the authors to add some discussion about the significance and novelty of this research.

Major comments:

1.      Can the authors present some data to show the enhanced permeability of nanoprodrug?

2.      It seems that result 2.5 should in vivo studies but not in vitro studies. (page9 line268)

Minor comments

Round 2

Reviewer 1 Report

First of all I would like to thank the authors for the extensive revision of he original manuscript. However, despite the evident changes, many improvements are still needed.

First of all there are too many typos, please check and correct them.

The english language is still inaccurate. In the following you can find some examples:

1) Line 100, "which are mainly" should be "which is mainly"

2) Line 107, "Then BODIPY 2 was click" should be "Then BODIPY 2 was clicked" and "containing an ROS" should be "containing a ROS"

3) Line 108-109, the meaning of the sentence is not clear, "substituting for 1O2 inert" is a non sense.

4) Line 272, "The results..." is another sentence that is not clear.

5) Line 420, "After removing..." is another sentence that is not clear.

6) Line 415, The heading number is not correct, it should be 3.4 (check also the following siìubsections)

Other important issues that should be addressed are:

a) Line 157, the authors mention DPBF. The extended name should be specified; moreover also the mechanism by which DPBF acts should be detailed for a better comprehension of the test.

b) Line 200, the authors mention a "redshift phenomena" that is not so evident from Figure 2c.

c) All the tests performed on cells have been carried out both on 4T1 and on MCF-7 cell lines, but only Figures referred to data from experimnts on 4T1 have been in the main text. I suggest to include in the main text also the Figures referred to MCF-7.

d) In the Supporting Information Section, Figure S8(a) shows a BODIPY emission for BTC NPs + BSO and no emission for BCC NPs. This is not correct, according to what reported in the text

e) The authors should reconsider the entire text in the materials and methods section. The way the experimental procedures are reported is so poorly clear, in many cases the sentences are too long and in some cases relevant details are missing. For example:

Line 396, the authors says that "The value was recorded", but it is not specified which value.

Line 400, the authors says that the fluorescence emission was measured, but there is no mention about the excitation wavelength and the emission wavelength.

Line 409, the authors say that the test was repeated 12 times, why?

Line 497, the sentence starting with "After the BTC NPs..." is too long and difficult to understand

As you can imagine there is still a lot of work to do, so I can reconsider the paper for further review after major revisions    

Round 3

Reviewer 1 Report

The manuscript has been improved according to the raised issues, I thank the authors for this; however there are still some revisions that are needed, especially many typos. The english editing must to be carefully revised.

In the following some suggestions:

Page 6 Line 150: “Figure 1. (a) Fluorescence emission spectra of ZnPc, 1, 2, BCC and BTC absorbed between 0.04-0.05 at 610 nm” what is 0.04-0.05? The authors should indicate the solvent, the concentration of the solutions and the excitation wavelength.

Page 7, Line 187: “Pretreatment with 2.5 μM of GSH for 120 min, the fluorescence intensity of CPT from BTC was very low under the same laser irradiation (Figure S6b).” This sentence is not clear, please rephrase.

So I will reconsider the manuscript for publication in the International Journal of Molecular Biology after major revisions.

Round 4

Reviewer 1 Report

The authors have addressed all the queries, so the manuscript can be published as is